# Predictive Coding, Variational Autoencoders, and Biological Connections

**Joseph Marino**
Computation & Neural Systems
California Institute of Technology
Pasadena, CA 91125
jmarino@caltech.edu

## Abstract

Predictive coding, within theoretical neuroscience, and variational autoencoders, within machine learning, both involve latent Gaussian models and variational inference. While these areas share a common origin, they have evolved largely independently. We outline connections and contrasts between these areas, using their relationships to identify new parallels between machine learning and neuroscience. We then discuss specific frontiers at this intersection: backpropagation, normalizing flows, and attention, with mutual benefits for both fields.

## 1 Introduction

Perception has been conventionally formulated as hierarchical feature detection [52], similar to discriminative deep networks [34]. In contrast, predictive coding [48, 14] and variational autoencoders (VAEs) [31, 51] frame perception as a generative process, modeling data observations to learn and infer aspects of the external environment. Specifically, both areas model observations, $\mathbf{x}$, using latent variables, $\mathbf{z}$, through a probabilistic model, $p_\theta(\mathbf{x}, \mathbf{z}) = p_\theta(\mathbf{x}|\mathbf{z})p_\theta(\mathbf{z})$. Both areas also use variational inference, introducing an approximate posterior, $q(\mathbf{z}|\mathbf{x})$, to infer $\mathbf{z}$ and learn the model parameters, $\theta$. These similarities are the result of a common origin, with Mumford [45], Dayan et al. [9], and others [46] formalizing earlier ideas [59, 38]. However, since their inception, these areas have developed largely independently. We explore their relationships (see also [58, 37]) and highlight opportunities for the transfer of ideas. In identifying these ties, we hope to strengthen this promising, close connection between neuroscience and machine learning, prompting further investigation.

## 2 Background

**Predictive Coding** Predictive coding [8] is a theory of thalamocortical function, in which the cortex constructs a probabilistic generative model of sensory inputs, using approximate inference to perform state estimation. Top-down neural projections convey predictions of lower-level activity, while bottom-up projections convert the prediction error at each level into an updated state estimate. Such models are often formulated with hierarchies of Gaussian distributions, with analytical non-linear (e.g. polynomial) functions parameterizing the generative mappings [48, 14]:

$$p_\theta(\mathbf{z}_\ell|\mathbf{z}_{\ell+1}) = \mathcal{N}(\mathbf{z}_\ell; \boldsymbol{\mu}_{\theta,\ell}(\mathbf{z}_{\ell+1}), \boldsymbol{\Sigma}_{p,\ell}), \qquad p_\theta(\mathbf{x}|\mathbf{z}_1) = \mathcal{N}(\mathbf{x}; \boldsymbol{\mu}_{\theta,\mathbf{x}}(\mathbf{z}_1), \boldsymbol{\Sigma}_\mathbf{x}). \qquad (1)$$

Variational inference is performed using gradient-based optimization on the mean of $q(\mathbf{z}|\mathbf{x}) = \mathcal{N}(\mathbf{z}_\ell; \boldsymbol{\mu}_{q,\ell}, \boldsymbol{\Sigma}_{q,\ell})$, yielding gradients which are linear combinations of (prediction) errors, e.g.

$$\nabla_{\boldsymbol{\mu}_{q,1}}\mathcal{L} = \mathbf{J}^\mathsf{T}\boldsymbol{\varepsilon}_\mathbf{x} - \boldsymbol{\varepsilon}_1, \qquad (2)$$

where $\mathcal{L}$ is the objective, $\mathbf{J} = \partial\boldsymbol{\mu}_{\theta,\mathbf{x}}/\partial\boldsymbol{\mu}_{q,1}$ is the Jacobian, and $\boldsymbol{\varepsilon}_\mathbf{x}$ and $\boldsymbol{\varepsilon}_1$ are weighted errors, i.e. $\boldsymbol{\varepsilon}_\mathbf{x} = \boldsymbol{\Sigma}_\mathbf{x}^{-1}(\mathbf{x} - \boldsymbol{\mu}_{\theta,\mathbf{x}})$. Parameter learning can also be performed using gradient-based optimization. We discuss connections between these models and neuroscience in Section 3.

Real Neurons & Hidden Units Workshop @ NeurIPS 2019.

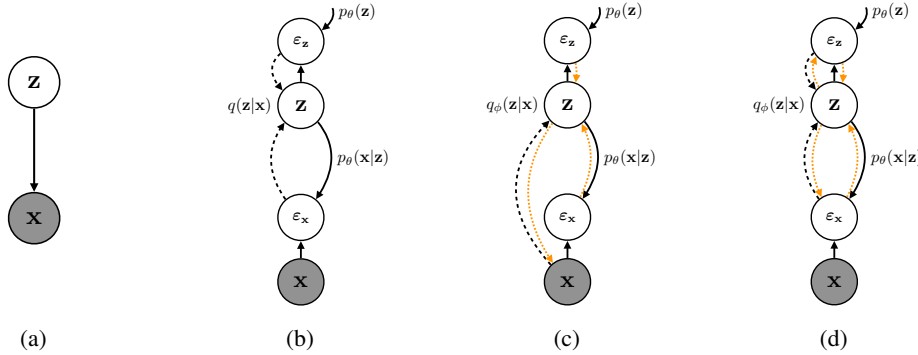

Figure 1: **Computation Graphs**. **(a)** Graphical model for the (single-level) latent variable model underlying both predictive coding and VAEs. **(b)** Computation graph for predictive coding. Black dashed lines denote inference computation (Eq. 2). **(c)** Computation graph for a standard VAE with direct amortized inference. Orange dashed lines denote parameter gradients, used for learning $\theta$ and $\phi$. **(d)** Computation graph for a VAE with an example of an iterative inference model [42].

**Variational Autoencoders**   VAEs are a class of Bayesian machine learning models, combining latent Gaussian models with deep neural networks. They consist of an *encoder* network with parameters $\phi$, parameterizing $q_\phi(\mathbf{z}|\mathbf{x})$, and a *decoder* network, parameterizing $p_\theta(\mathbf{x}|\mathbf{z})$. Thus, rather than performing gradient-based inference, VAEs *amortize* inference optimization with a learned network [19], improving computational efficiency. These networks can either take a *direct* form [9], e.g. $\boldsymbol{\mu}_q \leftarrow \text{NN}_\phi(\mathbf{x})$, or an *iterative* form [42, 41], e.g. $\boldsymbol{\mu}_q \leftarrow \text{NN}_\phi(\boldsymbol{\mu}_q, \nabla_{\boldsymbol{\mu}_q}\mathcal{L})$ or $\boldsymbol{\mu}_q \leftarrow \text{NN}_\phi(\boldsymbol{\mu}_q, \boldsymbol{\varepsilon}_\mathbf{x}, \boldsymbol{\varepsilon}_\mathbf{z})$, where $\text{NN}_\phi$ denotes a deep neural network. In both cases, gradients are obtained by reparameterizing stochastic samples, $\mathbf{z} \sim q(\mathbf{z}|\mathbf{x})$, separating stochastic and deterministic dependencies to enable differentiation through sampling [31, 51]. The parameters, $\theta$ and $\phi$, are learned using gradient-based optimization, with gradients calculated via backpropagation [54].

## 3   Connections, Contrasts, and Biological Correspondences

Predictive coding and VAEs are both conventionally formulated as hierarchical latent Gaussian models, with non-linear functions parameterizing the conditional dependencies between variables. In the case of predictive coding, these functions are often polynomials, whereas VAEs use deep networks, which are composed of layers of linear and non-linear operations. In predictive coding, Gaussian covariance matrices, e.g. $\boldsymbol{\Sigma}_\mathbf{x}$, have been treated as separate parameters, implemented as lateral weights between units at each level [14]. A similar, but more general, mechanism was independently developed for VAEs, known as normalizing flows [50, 30] (Section 4.2). Both areas have been extended to sequential models. In this setting, predictive coding tends to model dynamics explicitly, directly modeling orders of motion or *generalized coordinates* [15]. VAEs, in contrast, tend to rely on less rigid forms of dynamics, often using recurrent networks, e.g. [7], though some works have explored structured dynamics [26, 27, 40]. Both areas use gradient-based learning. In practice, however, learning in predictive coding tends to be minimal, while VAEs use learning extensively, scaling to large image and audio datasets, e.g. [49].

Predictive coding and VAEs both use variational inference, often setting $q(\mathbf{z}|\mathbf{x})$ as Gaussian at each level of latent variables. Predictive coding uses errors (Eq. 2) to perform gradient-based inference; note that this is a direct result of assuming Gaussian priors and conditional likelihood. In contrast, VAEs use amortized inference, learning to infer. This offers a solution to the so-called "weight transport" problem [36] for predictive coding; inference gradients require the Jacobian of the generative model (Eq. 2), which includes the transpose of generative weight matrices. Learning a separate set of inference weights avoids this problem, however, separate learning mechanisms are required for these inference weights.

The benefit of identifying these connections and contrasts is that they link neuroscience, through predictive coding and VAEs, to machine learning (and vice versa). While still under debate [2, 20],

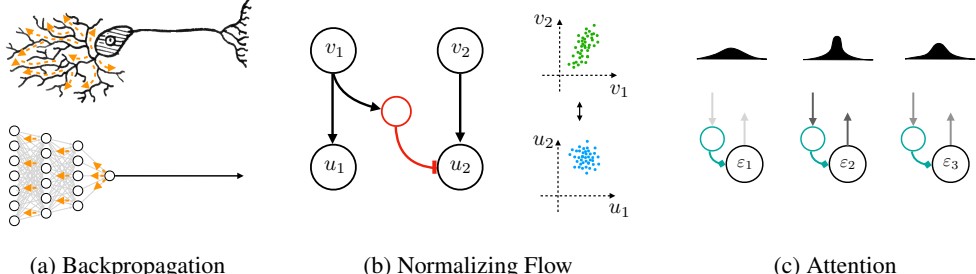

| (a) Backpropagation | (b) Normalizing Flow | (c) Attention |

Figure 2: **Frontiers**. **(a)** Comparing predictive coding and VAEs, deep networks are analogous to (ensembles of) dendrites. This suggests that similar mechanisms to backpropagation may operate *within* neurons. **(b)** Normalizing flows can be implemented with lateral inhibitory interactions. In the diagram, a correlated vector, $\mathbf{v}$, is decorrelated to a new vector, $\mathbf{u}$, simplifying the prediction space. **(c)** Attention mechanisms can be implemented using the precision (inverse variance) of predictions. Weighting prediction errors biases inference toward representing highly precise dimensions. In the diagram, various strengths of neuromodulation, corresponding to the precision of predictions, adjust the gain of error neurons.

biological correspondences of predictive coding have been proposed [3, 28]. Variable estimates and errors are hypothesized to be represented by neural activity (firing rate or membrane potential). These neurons would occur within the same cortical column, with variable neurons and error neurons in separate cortical layers. Interneurons could mediate inversion of errors and predictions, as well as lateral inhibition necessary for covariance matrices and attention (Section 4.3). Although many of the biological details remain unclear, one intriguing point emerges from this analysis: deep networks appear between variables, parameterizing the (non-linear) relationships between them. If variables represent neural activity, then deep networks are analogous to the mappings between neurons at separate levels in the hierarchy, i.e., dendrites. Thus, deep networks may more closely correspond with (a parallel ensemble of) dendrites, rather than entire networks of neurons. We discuss this implication and others in the following section.

## 4 Frontiers

### 4.1 Backpropagation & Learning

The biological correspondence of backpropagation [54] remains an open question. Backprop requires global information, whereas biology seems to rely largely on local learning rules [24, 43, 6]. A number of biologically-plausible formulations of backprop have been proposed [56, 33, 63, 25, 36], attempting to reconcile this disparity and others. However, recent formulations of learning in latent variable models [5, 35, 61] offer an alternative perspective: prediction errors at each level of the latent hierarchy provide a local signal [14], capable of driving learning of inference and generative weights.

In Section 3, we noted that deep networks appear between each latent level, suggesting a correspondence with dendrites rather than the traditional analogy as networks of neurons. This implies the following set-up: learning across the cortical hierarchy is handled via local errors at each level, whereas learning within the neurons at each level is mediated through a mechanism similar to backpropagation. Indeed, looking at the literature, we see ample evidence of non-linear dendritic computation [44] and backpropagating action potentials within neurons (Fig. 2a) [62, 32]. From this perspective, segmented dendrites [4, 22] for top-down and bottom-up inputs to pyramidal neurons could implement separate inference computations for errors at different levels (Eq. 2). While the mechanisms underlying these processes remain unclear, focusing efforts on formulating biologically plausible backpropagation from this perspective (and not supervised learning) could prove fruitful.

### 4.2 Normalizing Flows as Local Inhibition

We often consider factorized parametric distributions, as they enable efficient evaluation and sampling. However, simple distributions are limiting. *Normalizing flows* (NFs) [53, 10, 50] provide added complexity while maintaining tractable evaluation and sampling. They consist of a tractable *base*

distribution and one or more invertible *transforms*. With the base distribution as $p_\theta(\mathbf{u})$ and the transforms as $\mathbf{v} = f_\theta(\mathbf{u})$, the probability $p_\theta(\mathbf{v})$ is given by the *change of variables* formula:

$$p_\theta(\mathbf{v}) = p_\theta(\mathbf{u}) \left| \det \left( \frac{d\mathbf{v}}{d\mathbf{u}} \right) \right|^{-1}, \tag{3}$$

where $\det(\cdot)$ denotes matrix determinant and $|\cdot|$ denotes absolute value. The determinant term corrects for the local scaling of space when moving from $\mathbf{u}$ to $\mathbf{v}$. A popular family of transforms is that of autoregressive affine transforms [30, 47]. One example is given by

$$v_i = \alpha_\theta(\mathbf{v}_{<i}) + \beta_\theta(\mathbf{v}_{<i}) \cdot u_i, \tag{4}$$

where $v_i$ is the $i^{\text{th}}$ dimension of $\mathbf{v}$ and $\alpha_\theta$ and $\beta_\theta$ are functions. The inverse transform (Fig. 2b) is

$$u_i = \frac{v_i - \alpha_\theta(\mathbf{v}_{<i})}{\beta_\theta(\mathbf{v}_{<i})}, \tag{5}$$

a normalization (whitening) operation. Thus, we can sample from complex distributions by starting with simple distributions and applying local affine transforms. Conversely, we can evaluate inputs from complex distributions by applying normalization transforms, then evaluating in a simpler space.

Local inhibition is ubiquitous in neural systems, thought to implement normalization [13]. These circuits, modeled with subtractive and divisive operations (Eq. 5), give rise to decorrelation in retina [21], LGN [11], and cortex [29]. NFs offer a novel description of these circuits and agree with predictive coding. For instance, evaluating flow-based conditional likelihoods [1] involves whitening the observations, as performed by Rao & Ballard [48], to remove low-level spatial correlations. The same principle can be applied across time, where NFs resemble temporal derivatives [40], which are the basis of Friston's generalized coordinates [15]. Likewise, Friston's proposal [14] of implementing prior covariance matrices with lateral weights in cortex corresponds to a linear NF [30].

Local inhibition is also present in central pattern generator (CPG) circuits [39], giving rise to correlations in muscle activation. NFs are also being explored in the context of action selection in reinforcement learning [57, 60]. By providing a basis of correlated motor outputs, NFs improve action selection and learning, which can take place in a less correlated space that is easier to model. CPGs would likely be a form of *inverse* autoregressive flow [30] to maintain efficient sampling.

### 4.3   Attention via Precision Weighting

Predictive coding has proposed that prior covariance matrices, which weight prediction errors, could implement a form of *attention* (Fig. 2c) [55, 16]. Intuitively, decreasing the variance of a predicted variable pushes the model to more accurately infer and predict that variable. Biologically, this is hypothesized to be implemented via gain modulation of error-encoding neurons, mediated through neurotransmitters and synchronizing gamma oscillations [12]. This attentional control mechanism could bias a model toward representing task-relevant information. Deep latent variable models have largely ignored this functionality; when combined with active components, variances are typically held constant, e.g. [23]. Enabling this capacity for task-dependent perceptual modulation may prove useful or even essential in applying deep latent variable models to complex tasks.

## 5   Conclusion

We have identified commonalities between predictive coding and VAEs, discussing new frontiers resulting from this perspective. Reuniting these areas may strengthen the connection between neuroscience and machine learning. Further refining this connection could lead to mutual benefits: neuroscience can offer inspiration for investigation in machine learning, and machine learning can evaluate ideas on real-world datasets and environments. Indeed, despite some push back [17], if predictive coding and related theories [18] are to become validated descriptions of the brain and overcome their apparent generality, they will likely require the computational tools and ideas of modern machine learning to pin down and empirically compare design choices.

### Acknowledgments

We thank Sam Gershman for comments on this manuscript, and we thank Karl Friston for useful early discussions related to these ideas.

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
