# OpenReview forum: "Predictive Coding, Variational Autoencoders, and Biological Connections"
_NeurIPS.cc/2019/Workshop/Neuro_AI — Real Neurons & Hidden Units @ NeurIPS 2019 Poster_

### Official Review · AnonReviewer2 · 2019-09-24
**A brief review of predictive coding and variational autoencoders, with suggestions for future research**

**Clarity:** 2

**Comment:**

Strengths:

The ideas floated in this article have a lot of potential to launch a research topic. The structure is strong and would make for a good first draft of a research grant.

Areas for improvement:

The thesis needs to be more focused. What is(are) the research question(s) that you want the reader to reach by the time they finish reading? Narrowing this down and making it clear is an absolute must. In this vein, I felt the discussion of normalizing flows was particularly promising.

**Category:**

Common question to both AI & Neuro

**Clarity Comment:**

While the descriptions of the concepts in this article are clear, the overall synthesis and thesis of the article are not.

**Evaluation:**

2: Poor

**Importance:**

3: Important

**Importance Comment:**

Predictive coding is a current theory in systems neuroscience with a lot of potential for development by looking at deep generative models. Likewise, deep generative models inspired by thalamocortical architecture and dynamics could result in improvements to online perceptual learning.

**Intersection:**

4: High

**Intersection Comment:**

The article is addressing open questions relevant to both AI and neuroscience.

**Rigor Comment:**

Since there are no results in this submission, I have read it like a synthesis of two divergent literatures. The initial descriptions of predictive coding and variational autoencoders are precise and succinct. However, the comparisons and contrasts is very shallow. The discussion on biologically plausible backpropagation is worthwhile, however, the connection to either predictive coding or variational autoencoders is not made. The discussion on normalizing flows is interesting and links with predictive coding are established, so I accept from this article that this could be an interesting frontier for research.

**Technical Rigor:**

1: Not convincing

---

### Official Review · AnonReviewer1 · 2019-09-26
**Backpropagation and normalizing flows**

**Clarity:** 4

**Comment:**

The work is very clear to read and follow - overall, the presentation is strong. There are many potential areas of interest that arise from the ideas outlined here.

Overall, however, the work would benefit from more discussion by the authors of why they chose these topics (i.e. which of these ideas is of particular interest, such that they think that these are an interesting new research direction). Some sort of brief outlook or summary for future consideration would be of added value at the end of the document.

**Category:**

Common question to both AI & Neuro

**Clarity Comment:**

The overall text was well-written and easy to follow. The figures were only somewhat helpful, but as this is a synthesis paper, added to the overall story.

**Evaluation:**

3: Good

**Importance:**

3: Important

**Importance Comment:**

Predictive coding remains of great interest in systems neuroscience - with much effort devoted to linking theory to biological function. Thalamocortical architecture has been relatively well characterized biologically, suggesting it may be a good architecture for future efforts.

**Intersection:**

3: Medium

**Intersection Comment:**

Backpropagation is an area of great interest for both AI and neuroscience; in this sense, this paper highlights interesting ways in which this could be a future research direction. Broadly, I wonder at statements of biology relying on local learning rules - of course it does, but the studies referenced are largely in neuronal culture dishes. Perhaps by understanding dynamics at a systems scale (in small model organisms perhaps), it may be both local and global. It is unclear based on the authors' framing if their deep network approach allows for such flexibility.

**Rigor Comment:**

As the authors' goal seems to have been to present a synthesis of ideas from the field, the technical rigor may be acceptable on these grounds.

**Technical Rigor:**

2: Marginally convincing

---

### Official Review · AnonReviewer3 · 2019-09-26
**Very preliminary work connecting two related paradigms, but providing only very speculative new links**

**Clarity:** 3

**Comment:**

The paper provides a high-level overview of predictive coding and VAEs and speculatively connects these two methods to outstanding questions in neuroscience (for example: to the function of lateral connections and the question of whether backpropagation-like computations occur in the brain). This work is very preliminary and presents no technical results.

**Category:**

Common question to both AI & Neuro

**Clarity Comment:**

The exposition is fairly clear.

**Evaluation:**

1: Very poor

**Importance:**

2: Marginally important

**Importance Comment:**

The paper provides a high-level overview of predictive coding and VAEs and speculatively connects these two methods to outstanding questions in neuroscience (the function of lateral connections and whether backpropagation occurs in the brain).

**Intersection:**

5: Outstanding

**Intersection Comment:**

This paper attempts to build a bridge between variational autoencoders (an important framework for generative modeling and unsupervised learning in ML) and predictive coding (a controversial, but potentially powerful explanatory framework in neuroscience).

**Rigor Comment:**

The high-level overview of VAEs and predictive coding appears to be correct. However, the connections made in this paper to neuroscience (in the sections on backpropagation and normalizing flows) are largely speculative. No substantive predictions are made, and the biological details are not examined with enough granularity to draw any solid conclusions. For example, it's unclear in what sense normalizing flows may "help justify design choices in predictive coding," as claimed.

**Technical Rigor:**

2: Marginally convincing

---

### Decision · Program_Chairs · 2019-10-02

Accept (Poster)